# Are BPA Substitutes as Obesogenic as BPA?

**DOI:** 10.3390/ijms23084238

**Published:** 2022-04-11

**Authors:** Fabiana Oliviero, Alice Marmugi, Catherine Viguié, Véronique Gayrard, Nicole Picard-Hagen, Laila Mselli-Lakhal

**Affiliations:** ToxAlim (Research Centre in Food Toxicology), Université de Toulouse, INRA, ENVT, INP-Purpan, UPS, 31027 Toulouse, France; fabiana.oliviero@hotmail.it (F.O.); alice.marmugi@bayer.com (A.M.); catherine.viguie@inrae.fr (C.V.); veronique.gayrard@envt.fr (V.G.); nicole.hagen@envt.fr (N.P.-H.)

**Keywords:** BPA substitutes, metabolic disorders, endocrine disruptors

## Abstract

Metabolic diseases, such as obesity, Type II diabetes and hepatic steatosis, are a significant public health concern affecting more than half a billion people worldwide. The prevalence of these diseases is constantly increasing in developed countries, affecting all age groups. The pathogenesis of metabolic diseases is complex and multifactorial. Inducer factors can either be genetic or linked to a sedentary lifestyle and/or consumption of high-fat and sugar diets. In 2002, a new concept of “environmental obesogens” emerged, suggesting that environmental chemicals could play an active role in the etiology of obesity. Bisphenol A (BPA), a xenoestrogen widely used in the plastic food packaging industry has been shown to affect many physiological functions and has been linked to reproductive, endocrine and metabolic disorders and cancer. Therefore, the widespread use of BPA during the last 30 years could have contributed to the increased incidence of metabolic diseases. BPA was banned in baby bottles in Canada in 2008 and in all food-oriented packaging in France from 1 January 2015. Since the BPA ban, substitutes with a similar structure and properties have been used by industrials even though their toxic potential is unknown. Bisphenol S has mainly replaced BPA in consumer products as reflected by the almost ubiquitous human exposure to this contaminant. This review focuses on the metabolic effects and targets of BPA and recent data, which suggest comparable effects of the structural analogs used as substitutes.

## 1. Introduction

Numerous experimental, clinical and epidemiological studies suggest that exposure to environmental contaminants can disrupt endocrine and metabolic functions and contribute to the development of obesity and associated metabolic disorders, such as Type 2 diabetes, coronary heart disease and hypertension [1]. According to this new concept, environmental contaminants would play the role of environmental obesogens. Many epidemiological studies established a positive correlation between exposure to Bisphenol A (BPA), phthalates, pesticides, alkylphenols and the prevalence of cardiovascular diseases, diabetes, and weight gain [2]. The most studied environmental obesogen is BPA. BPA was identified in the 1930s as a synthetic estrogen that had a potential impact on female reproductive function. However, BPA was not used as such because of the discovery of a more potent estrogenomimetic compound: diethylstilbestrol (DES). The use of DES to prevent miscarriage was then revealed to be disastrous for millions of people who developed genital anomalies, sterility and an acute risk of cancer following in utero exposure [3]. In 1960, BPA began to be extensively used for the industrial manufacture of polycarbonate plastics and epoxy resins. Thereby, BPA can be found in the inner coatings of tins, beverage cans and much food packaging. In addition, it is found in CDs, DVDs, some electronic devices, mobile phones, glasses, contact lenses and thermic ink receipts. Epoxy resins containing BPA are also used for water storage, transportation systems and some dentary cements [4]. Frederick Vom Saal, a biologist and professor at the Missouri University in Columbia (United States) was the first to report on the effects of low doses of BPA on the reproduction system of male mice born to mothers exposed to BPA [5]. These results were then largely supported by many studies that revealed the effects of BPA on the reproduction system, immune system and energy metabolism [6]. Numerous epidemiological and experimental studies focused on BPA as a metabolic disruptor. They showed that BPA could exert effects on all organs involved in the regulation of energy homeostasis, such as adipose tissue, pancreas, liver, muscle and brain [7]. Exposure to low doses of BPA in adults or in the perinatal period was associated with weight gain, the disruption of carbohydrate and lipid homeostasis and an effect on brain regions involved in food intake [7]. These and other studies have led to the ban of BPA in some countries. It was first banned in baby bottles in Canada in 2008 and in all food-oriented materials in France from 1 January 2015. More recently, in 2016, BPA was identified as a substance of very high concern, that is to say particularly dangerous, by the Committee of the Member States of the European Chemicals Agency because of its reprotoxic properties, and in 2017 for its endocrine disrupting properties for human health and the environment. It has since been replaced by other compounds from the bisphenol family such as BPS, BPF and BPAF. Most of those substitutes were selected based on their stability properties despite a very poor toxicological evaluation. Since BPA substitutes are structurally similar to BPA, it is expected that they may also have the same obesogenic effect. This review summarizes the results of studies supporting the metabolic disruptive effects reported for BPA and investigates whether the same obesogenic properties are reported for BPA analogs.

## 2. A Strong Link between BPA and Metabolic Disorders

A very large number of studies are devoted to the effect of BPA on the development of metabolic disorders. A recent meta-analysis conducted on 133 studies carried out in humans and selected according to exposure relevance revealed an association between exposure to BPA and a higher risk of developing Type II diabetes [8]. Urinary and plasmatic levels of BPA are positively associated with a higher risk of developing Type II diabetes. A study based on data from the National Health and Nutrition Examination survey (NHANES), including 1521 participants, also revealed higher BPA concentrations in obese participants, therefore suggesting an association between BPA and obesity [9]. An association also exists between exposure to BPA and acute insulin resistance, general obesity, abdominal obesity and the prevalence of diabetes [10]. Numerous epidemiological studies strengthen these observations [11,12,13].

These epidemiological data are supported by experimental data mainly in rodents showing an effect of BPA on organs involved in energy metabolism such as the liver, skeletal muscle, adipose tissue, pancreas and central nervous system (Figure 1). These studies have highlighted a number of characteristics of BPA, such as low-dose effects on adipocyte differentiation and on insulin production by β-pancreatic cells [14]. The effects are observed during adult exposure as well as after the perinatal period, which represents a more sensitive window of exposure [15,16,17].

### 2.1. Effect of BPA on Body Weight

In rodents, maternal exposure to BPA was shown to increase postnatal body weights [16,18,19,20]. The dose–response relationship between BPA exposure and body weight gain often follows a non-monotonic inverted-U shape effect with an increase in body and fat mass in response to low doses (below the NOAEL) that were not always observed at high doses [16,19,20]. These non-monotonic effects are not always seen in both sexes. In females exposed in utero, adipose tissue mass is increased at low doses of BPA (0.26 mg/kg/j) but not at higher doses (2.72 mg/kg/j). In males, adipose tissue mass is increased proportionally to BPA exposure dose [15]. Body weight increase is often more pronounced and persistent in female offspring. This sexual dimorphism is not seen in all experimental conditions. In the study by Wei et al., an increased body weight of rats exposed in utero to BPA is observed, independently of sex, in standard feeding conditions and under a high-fat or carbohydrate diet [21]. Studies conducted on adults revealed that the exposure of gestating mice to BPA (100 µg/kg) leads to an increased body weight [22]. Some studies revealed a decreased weight following perinatal exposure to BPA and others revealed no effects on body weight [23,24,25]. The differences obtained by the different studies mentioned above could be explained by the strains used, differing from one study to another; some strains were more sensitive to estrogenomimetic processes that mediate at least in part the effect of BPA on energy hoemostasis [26]. In addition, the exposure window, duration and mode of administration of BPA [26], and the type of feed [27] are key factors to take into consideration. Therefore, the impact of BPA on body weight gain can differ according to experiments. However, observations on key organs of energetic metabolism (liver, skeletal muscle, adipose tissue and pancreas) support the fact that BPA is not only an endocrine disruptor but also a metabolic disruptor. Since the route of exposure to BPA is mainly through food and beverage containers, the more an individual consumes processed foods stored in plastic containers, the more they will be exposed to BPA.

### 2.2. Effect of BPA on the Central Nervous Functions Related to Energy Homeostasis

Proopiomelanocortin (POMC) neurons in the hypothalamic arcuate nucleus (ARC) are anorexigenic neurons that inhibit food intake and increase metabolic rate, while agouti-related peptide (AgRP) and neuropeptide Y (NPY) are orexigenic neuropeptides that stimulate appetite and reduce metabolic rate. These two sets of neurons form the hypothalamic melanocortin system, the physiological system that regulates feeding and energy balance. The activity of the melanocortin system is controlled by hormones, such as estradiol, leptin, ghrelin, and is sexually dimorphic. MacKay et al. analyzed whether in utero BPA exposure could alter the development of the melanocortin system and be linked to the obesogenic effect of BPA [28]. This study revealed impaired glucose tolerance in males exposed to BPA associated with reduced POMC neuron innervation. This effect was associated with increased NPY and AgRP expression in ARC when mice were fed with HFD. In females, BPA exposure induced increased body weight gain, food intake, adiposity and leptin concentrations, associated with reduced POMC mRNA expression in the ARC when fed an HFD diet. In BPA-exposed females, estrogen receptor α presents similar patterns of expression than in males, suggesting a masculinizing effect of BPA. This study demonstrates that in utero exposure to BPA alters the structure of the hypothalamic energy balance system and increases vulnerability to developing metabolic disorders. In 2017, the same authors extended their study to determine whether their prior observations were simply consequences of obesity or a phenotype produced by BPA exposure [29]. Therefore, they studied leptin sensitivity and hypothalamic structures in BPA-exposed animals before the onset of obesity or metabolic phenotypes. BPA-exposed animals were resistant to leptin-induced suppression of food intake, body weight loss, and hypothalamic POMC upregulation. Both males and females had a reduced density of POMC projections in the paraventricular nucleus of the hypothalamus. These results suggest that BPA may exert its effects through developmental programming of the melanocortin system, permanently altering the neurobiology of metabolic homeostasis. Salehi et al. explored whether the effect of BPA on POMC neurons was direct by using different cell lines, including POMC-expressing cell models [30]. This study demonstrated that exposure to BPA significantly induced the mRNA levels of POMC in primary cultures and cell lines. Furthermore, cell treatments with anti-inflammatory compounds, or with a PPARγ antagonist, abolished BPA-mediated POMC induction, indicating that BPA may have direct effects on hypothalamic POMC neurons through neuro-inflammatory mechanisms and PPARγ receptor.

### 2.3. BPA, a Disruptor of Carbohydrate Homeostasis

Many studies, mainly conducted by Angel Nadal’s team, showed that exposure to BPA leads to the dysregulation of carbohydrate metabolism by a mechanism involving ostrogen receptors in Langherans islets [22,31,32,33,34,35]. In adult male mice, a few days of subcutaneaous exposure (1 and 4 days) to low doses of BPA (10 and 100 µg/kg/day) induces an alteration of glucose tolerance, hyper insulinemia, and increased the content of insulin in β-cells [33,36]. The same effect was observed in vitro in the presence of BPA at concentrations of 1 nM and 10 nM [36]. Langherans islets of adult mice orally exposed to 100 µg/kg/day of BPA present increased insulin secretion in response to glucose (Figure 2) [37].

Mice exposed to BPA (10 µg/kg/day) during gestation period also developed persistent hyperinsulinemia. The offspring presented with a predisposition to metabolic syndrome development at adulthood (insulin resistance, alteration of insulin secretion and calcic signaling in β-cells) [22]. Similar results were observed in HFD-fed rats, which present earlier and exacerbated effects, mainly at low doses of BPA (50 µg/kg/day) [21]. Moreover, β-cells of exposed animals present structural defects. Mitochondria and rough endoplasmic reticulum are hypertrophied. The proportion of mature secretory granules is decreased in animals exposed to BPA fed with a standard diet and almost absent under a high-fat diet. In mice exposed to BPA and fed a high-fat diet, islets are disorganized and cells undergo pycnose (irreversible condensation of chromatin leading to necrosis of cells) [21]. In addition, chronic exposure of β-cells (TC-6) to BPA modifies the expression of key proteins involved in endoplasmic reticulum stress response [38]. Unlike many observations suggesting that BPA is a weak agonist of estrogen receptors [39], Alonso-Magdalena et al. revealed that BPA (1 nM range) could similarly mimic estradiol (E2) effects in β-cells [33]. Non-genomic ERα is involved in the long-term effects of BPA by increasing gene transcription of insulin precursor via ERK1/2 phosphorylation [36], whereas membrane ERβ is involved in pulsatile activity of insulin. Low doses of BPA (1 nM) rapidly decrease the activity of KATP channels via Erβ, which depolarizes the membrane and increases intracellular calcium levels and, therefore, induces insulin secretion [32,37]. Other nuclear receptors could be involved, such as transmembrane domains receptors (RCPG). The RCPG GPR30/GPER, a target of BPA [40] has recently been identified as a mediator of the effects of insulin in response to E2 [41].

BPA could also affect pancreatic α-cells. The exposure of α-cells to BPA (1 nM) mimics E2 effect by blocking the Ca^2+^ effect involved in glucagon release. These effects could occur via estrogen membrane receptors and involve G proteins that activate nitric oxide synthase (NOS) and cGMP-dependent protein kinase [42].

These data suggest that, in the long run, exposure to BPA could be detrimental for β and α cell function and, therefore, be an important factor in the etiology of type II diabetes and development of insulin resistance.

### 2.4. BPA, a Disruptor of Lipid Metabolism

BPA has been shown in several experimental setups to alter lipid regulation in different tissues, particularly by interfering with insulin-mediated pathways

#### 2.4.1. Effect of BPA on Adipose Tissue

Obesity results from increased white adipose tissue mass due to an increased size and number of adipocytes. The use of various cell lines revealed an effect of environmental endocrine disrupters (4-nonylphenol, tributyltin…) in white adipose tissue. In humans, BPA serum concentrations are more important in obese patients than in normal patients [43]. BPA is detected in human white adipose tissues with a concentration of 3.16 ± 4.11 ng/g of tissue [44].

Numerous studies carried out in vivo on rodents or in vitro on human adipocytes have shown an adipogenic effect of BPA [45,46,47,48]. In vitro experiments have shown that BPA (2 to 20 µg/mL) induces adipocyte differentiation by increasing triglyceride content and LPL (lipoprotein lipase) activity in the presence of insulin [48,49]. BPA increases the gene expression of adipogenic transcription factors (C/EBPβ: alpha CAAT enhancer binding protein, PPARγ and FAS) in 3T3-L1 pre-adipocytes [46,47] and adipose-derived mesenchymal stem cells [45]. BPA increases glucose uptake in basal conditions and in response to insulin. This could partly be explained by an increased synthesis of glucose transporter GLUT4 [50]. On the contrary, exposure of human stem cells to BPA decreases LPL activity and triglyceride accumulation [51].

White adipose tissue is not only an energy storage organ but also an endocrine gland that secretes different metabolically active peptides with regulatory properties called adipokines. BPA affects the production and secretion of adiponectin by 3T3-L1 adipocytes [52] and by human adipose tissue explants [53], mostly following a non-monotone dose response [53,54].

Studies conducted on mice revealed that female offspring, in utero exposed to BPA and fed a high-fat and carbohydrate diet, present with elevated levels of leptin, particularly at low doses of BPA (0.3 mg/k/day), but this effect is not found in males [15]. In addition, female rats exposed to BPA in utero (70 µg/kg/day) present an excessive white adipose tissue mass associated with adipocyte hypertrophy and the upregulation of lipogenic genes, such as C/EBP-alpha, PPARγ, SREBP-1c, LPL, FAS and SCD-1. In utero exposed male rats seem less affected but present negative effects when fed a high-fat diet [20]. In the same way, perinatal exposure of rats to 50 µg/kg/day of BPA leads to adipocyte hypertrophy under standard feeding conditions and under a high-fat diet, independent of sex [21].

#### 2.4.2. Effect of BPA on the Liver

Epidemiological studies link exposure to BPA to the occurrence of non-alcoholic fatty liver disease (NAFLD), which is considered a predominant chronic liver disease worldwide and a component of metabolic syndrome [55,56,57,58]. These studies are supported by experimental data showing that BPA deregulates many energy metabolic pathways in the liver [20,59,60,61,62].

Female rats exposed in utero to 70 µg/kg/day of BPA present an increased expression of hepatic lipogenic factors (SREBP-1c, FAS et ACC) [20]. In parallel, in response to low doses of BPA (10–12 to 10–6 M), human HepG2 hepatocytes present an accumulation of intracellular lipids [62]. Studies conducted by Marmugi et al. revealed an accumulation of hepatic lipids in mice exposed to BPA via food (50, 500 and 5000 µg/kg/day). This lipid accumulation was associated with hyperinsulinemia and the disruption of genes involved in hepatic lipogenesis and cholesterogenesis [61]. These effects on hepatic lipid metabolism were confirmed by many studies that followed, whether they studied exposures in adults or in perinatal mice [59,60,63,64].

All of the effects described by Marmugi et al. follow a non-monotonic dose–response effect, as illustrated in Figure 3 [61]. BPA exposure was also shown to deregulate cholesterol metabolism after in utero and adult exposure [64,65].

High doses of BPA (50 mg/kg/day) induce the production of reactive oxygen species and the repression of anti-oxidant genes (catalase glutathione reductase, glutathione transferase, glutathione peroxidase) leading to hepatotoxicity in rats [66]. Acute exposure to BPA at a dose lower than the NOAEL (1.2 mg/kg/day) can induce hepatic lesions and mitochondrial dysfunctions by increasing oxidative stress, inflammation and lipid peroxidation in mice [67]. In addition, ALAT and ASAT transaminases, which are markers of hepatic injury and serum levels of inflammatory cytokines (IL-6 and TNFα), were shown to be strongly increased 6 h after BPA injection. Studies conducted on human hepatocyte cell lines (HepG2) confirm these results; a short exposure to 10 or 100 nM of BPA induces structural and functional alterations in mitochondria (decreased oxygen consumption, ATP production and membrane permeability) associated with increased oxidative stress and inflammation [62,67]. BPA induces endoplasmic reticulum stress via the production of endoplasmic reticulum oxidoreduction (ERO) in hepatic macrophages [68]. The chronic activation of endoplasmic reticulum stress plays an important role in the development of insulin resistance and obesity [69]. The analysis of hepatic signaling pathways of gestating mice reveals that the Akt phosphorylation pathway (Tht308 residue) is decreased following BPA exposure (10 µg/kg/day), which reflects insulin resistance [22]. In male mice, BPA treatment leads to the suppression of the IRS-1 protein, but the Akt pathway does not seem to be affected [70].

#### 2.4.3. Effect of BPA on the Muscle

In skeletal muscle and in physiological conditions, insulin induces Akt phosphorylation. This response is completely suppressed in gestating mice exposed to BPA (10 µg/kg/day), reflecting insulin resistance [22]. BPA treatment can also affect the MAPK signaling pathway as revealed by the inability of insulin to induce ERK phosphorylation [70].

### 2.5. Mechanism of BPA Action

The effect of BPA on the homeostasis of energy metabolism could be linked to its interaction with several nuclear receptors.

BPA is a weak agonist of ERα and ERβ estrogen receptors and its estrogen mimetic properties have long been considered responsible for its effects. However, BPA presents an affinity for ERα and ERβ receptors thousands of times less than estradiol [71]. Many studies reveal that BPA interacts with other receptors, suggesting that its estrogenic mimetic properties cannot explain all adverse effects of BPA [71].

Several studies also demonstrated that BPA binds to the androgen nuclear receptor (AR) [72,73]. AR is mainly expressed in the testicles, prostate, adrenal glands, kidneys, brain and pituitary gland. Unlike ERs, BPA antagonizes AR and its affinity is in the micro-molar range [72]. Low-BPA-dose effects could partly be explained by synergistic actions through ER receptors (agonist and feminizing actions) and AR receptors (antagonist of the masculinizing effect).

BPA is a potent ligand of ERRγ nuclear receptor (estrogen-related receptor gamma) [74,75,76]. The fact that this receptor binds to promoters of estrogen receptor target genes suggests its involvement in BPA endocrine disrupter effects [77]. ERRγ is known for a positive regulation of adipocyte differentiation [78] the mediation of glucagon effects on liver, induction of genes involved in gluconeogenesis (Pepck, G6Pase), increase in glucose production and disruption of hepatic insulin signaling [79]. Therefore, ERRγ could contribute to the effects of BPA on energetic metabolism.

Watson et al. suggested that BPA could exert part of its effects through membrane ERα and ERβ [80,81]. Membrane localization of these receptors could result from post-translational modifications such as palmitoylation [82], which could explain some rapid effects of BPA. However, this mechanism of action cannot explain low-dose effects. In fact, it is assumed that membrane forms present the same affinity for BPA as nuclear forms.

A second mediator of the non-genomic effects of BPA could be the G protein transmembrane receptor GPR30. In its active state, GPR30 activates a G protein triggering a signaling cascade. During extended exposure to ligands, receptors are internalized and desensitized leading to stoppage of signal transduction. GPR30 is localized in endoplasmic reticulum and can bind to low doses of BPA [83,84]. GPR30 is involved in the insulinotropic effects of E2 in β pancreatic cells suggesting a mode of action similar to BPA [35].

Xenosensors CAR and PXR are also BPA targets. BPA is described as an agonist of PXR human form [85,86,87] but not of the murine form. It is suggested that specific residues of the human PXR binding pocket allow for BPA binding [86].

## 3. Are BPA Substitutes as Obesogenic as BPA?

While BPA is banned from food packaging, many questions remain regarding the isks presented by BPA substitutes, in particular bisphenol S (BPS) and bisphenol F (BPF), which are authorized by regulations. A study revealed that, in several countries (Japan, USA, China, Kuwait and Vietnam), BPS was detected in urine at concentrations comparable to those of BPA [88]. Many epidemiological studies report an association between BPA or BPF exposure and obesity or diabetes. In an analysis of the NHANES, urinary BPS was positively associated with general obesity, especially in children and teenagers [89,90], and urinary 4,4-BPF concentrations were elevated in obese teens [90]. Similarly, urinary BPS concentrations were reportedly associated with a significantly increased risk of type 2 diabetes, [91]. However, other studies did not find any association between BPS or BPF exposure and hyperglycemia [92], or insulin resistance [93]. These epidemiological studies are supported by experimental studies showing that BPA analogs may have an impact on human health, especially in terms of obesity and other adverse health effects [94].

### 3.1. Bisphenol S Effects

As a substitute of BPA, BPS has replaced this substance in several food packaging products, thermal papers, paper products, personal care products and various other industrial applications. BPS as BPA is mostly metabolized by conjugation reactions [95]. A recent study carried out in piglets has shown that the amount of ingested BPS that reaches the general bloodstream is about 100 times higher than that of BPA [96]. This finding of a much higher oral bioavailability of BPS compared to BPA (57% vs. 0.5%) has been recently confirmed by the high estimate of BPS oral bioavailability in humans (62%) [95]. The much higher BPS oral bioavailability, combined with its longer persistence [97], explains the much higher systemic exposure to active BPS than BPA [98]. In addition, BPS is a hormonally active substance that displays estrogenic activities comparable to those of BPA [99]. Therefore, these results suggest that the replacement of BPA by BPS may lead to increased internal exposure to an endocrine-active compounds. Aside from its estrogen-like activities, many studies revealed that BPS displays metabolic effects similar to those of BPA.

#### 3.1.1. BPS Presents the Same Adipogenic Effect as BPA

Numerous studies show that BPS has the same adipogenic properties as BPA [100,101,102,103,104]. Boucher et al. compared the capacity of BPA and BPS to induce adipocyte differentiation [103]. The authors exposed mouse pre-adipocyte cell lines to different BPA and BPS concentrations and analyzed its adipogenic effects by evaluating the lipid accumulation and gene expression of adipogenesis markers. This study revealed that BPS, as with BPA, induced lipid accumulation and increased adipogenic gene expressions, such as lipoprotein lipase and adipocytary protein 2 and that this effect involves the PPARγ nuclear receptor. The potency of BPS adipogenic effects were even greater than those of BPA. This greater adipogenic effect of BPS has also been reported by other authors [103]. This adipogenic effect was also observed in sheep and mice after gestational exposure [100,104]. Moreover, the same effect was revealed in human primary pre-adipocytes in cultures [101] with a differentiation of human primary pre-adipocytes exposed to BPS and an upregulation of the gene and protein expression involved in adipogenesis and lipid accumulation. The authors of the study suggest an involvement of ERα and PPARγ receptors.

#### 3.1.2. Effect of BPS on Carbohydrate and Lipid Metabolism

Perinatal BPS exposure studies have shown metabolic disorders in offspring [105,106]. Brulport et al. demonstrated that perinatal exposure to BPS significantly increased body weight, the weights of liver and epididymal white adipose tissue (epiWAT) [105]. A histopathological analysis showed that lipids were significantly accumulated in liver tissues and epiWAT with BPS exposure. Expressions of genes involved in the inflammatory pathways were significantly increased in liver tissues and epiWAT. A serum metabolomics study showed significant changes in the contents of metabolites associated with lipid and glucose metabolism [105]. Ivry-Del Moral et al. revealed an effect of BPS on lipid homeostasis following the perinatal exposure of C57Bl/6 mice [106]. Following perinatal exposure to BPS, offspring under a high-fat diet presented a more severe obesity than control mice with more important hyper-insulinemia and fat mass. Perinatal exposure to BPS also increased the plasmatic clearance of triglycerides in offspring, which revealed more plasmatic lipid storage. Twenty eight days of exposure to BPS induced an increased and fasted glycaemia and the induction of hepatic gluconeogenesis and glycogenolysis. Similar to BPA, BPS has been shown to activate the PPARγ receptor pathway in macrophages and significantly induce the expression of lipid-metabolism-related genes, including fatty-acid-binding protein 4 (FABP4) and cluster of differentiation 36 (CD36) [107]. BPS also disrupted glucose metabolism [108] and was associated with increased food consumption and body weight gain in mice [109]. Angel Nadal’s group also demonstrated that BPS, similar to BPA, increases glucose-induced insulin release by pancreatic β-cells. They evidenced a rapid response due to the closure of KATP channels and a long-term response via the regulation of ion channel gene expression [110].

#### 3.1.3. Effect of BPS on Central Nervous System

Several studies focused on the effect of BPA on the brain and behavior, but only one focused on the regions involved in the regulation of energy metabolism [109]. In 2018, Rezg et al. studied the effects of BPS on hypothalamic neuropeptides and feeding behavior [109]. They administered BPS to mice in drinking water for 10 weeks at 3 doses (25, 50, 100 µg/kg) and revealed an alteration in the mRNA levels of orexigenic hypothalamic neuropeptide (AgRP), which regulated feeding behavior and a dysregulation of the hypothalamic apelinergic system. These disruptions could lead to increased food intake and body weight. BPS exposure could, therefore, contribute to the development of metabolic disorders.

### 3.2. Other Bisphenols

While BPS is the most studied BPA substitute, other bisphenols, such as BPF, BPAF and BPB, are increasingly used in several industrial applications and are questionable in terms of safety. An epidemiological study established a link between urinary concentrations of bisphenol AF (BPAF) and type II diabetes [91]. BPA substitutes, including BPS, BPB, BPF, and BPAF, were shown to disrupt metabolic functions and insulin signaling in adipocytes under low, environmentally relevant concentrations through the inhibition of the PPARγ pathway [102]. Kidani et al. demonstrated that BPB, BPE and BPF decreased the amounts of intracellular and medium adiponectin [52]. A study conducted on zebra fish highlighted that the treatment of different doses of BPF induces increased gluconeogenesis and suppresses glycolysis [111]. Furthermore, BPF treatment reduces the gene and protein expression of insulin and gene expression of insulin receptor, suggesting a decreased insulin sensitivity. A study carried out on humans suggested an obesogenic effect of bisphenol F accumulation in brain [112]. This study highlighted an association between BPF accumulation in the hypothalamus and a more important incidence of obesity evaluated by body mass index. Some studies also showed that BPB activated nuclear receptors involved in the regulation of energy metabolism, such as PXR [113]; the activation of hPXR was dose-dependent, and BPB was more potent than BPA, as were hPXR agonists, at a low concentration (5 μM), and had comparable agonistic effects at high concentrations (10 and 25 μM) [113].

## 4. Conclusions

In conclusion, exposure to BPA is associated with the development of metabolic disorders such as obesity, type II diabetes or fatty liver disease. This link is attested by numerous epidemiological and experimental reports and reinforced by fairly extensive mechanistic studies. The studies carried out up to now aimed to assess the effects of exposure to BPA at doses corresponding to the ADI or the NOAEL; they will now have to integrate the doses corresponding to the actual human exposure to these compounds. The molecules used as substitutes for BPA have very similar chemical structures as BPA and could, therefore, present the same deleterious effects. In fact, the literature data reported on some of these substitutes reveal the same deleterious effects than BPA. Regarding the obesogenic potential of BPS, which is the most common BPA substitute, it appears that it could be a metabolic disruptor targeting several metabolic organs, both centrally and peripherally (liver, adipose tissue, muscle, central nervous system). The parameters affecting the gravity of the outcomes include sex (males are more susceptible than females), periods and duration of exposure and nutritional context (effects are more often observed in animals fed a high-fat diet). Metabolic disruptions may include body weight gain but also the disruption of lipid metabolism and altered food intake/behavior. Modes of action of BPS are not clearly defined. The disruption of plasmatic levels of estradiol and testosterone and expression levels of estrogen and glucocorticoid receptors, as well as mitochondrial dysfunction in liver have been reported following BPS exposure. In addition, recent data revealed that BPS was more bioavailable than BPA and a urinary analysis carried out in the United States, Japan or China showed that BPS was already detected in most of the population (more than 80% in the United States). BPA substitutes must, therefore, be carefully studied so as not to have similar structures and deleterious effects as BPA.

## Figures and Tables

**Figure 1 ijms-23-04238-f001:**
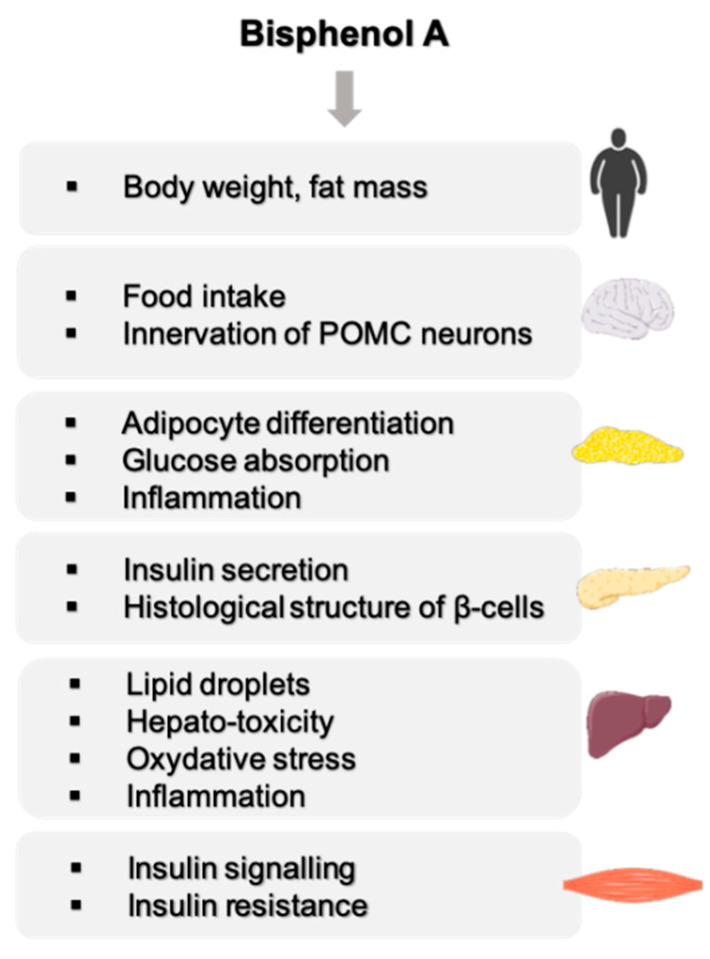
Overview of BPA effects on energy metabolism.

**Figure 2 ijms-23-04238-f002:**
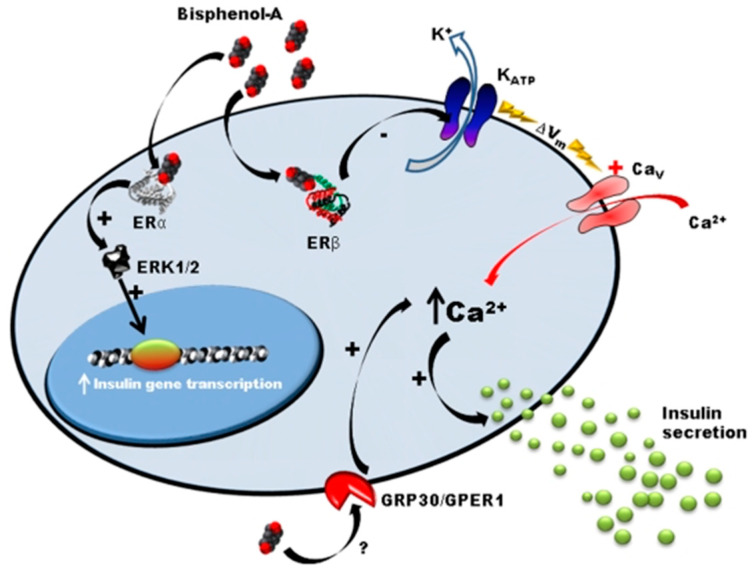
Mode of action of BPA on pancreatic β-cells [37]. This figure reports the mode of action of BPA on pancreatic cells. Low concentrations of BPA interact with Erα, Erβ and GPR30 receptors. ERα is involved in the regulation of pancreatic insulin biosynthesis in response to BPA. Erβ participates in the insulinotropic effect of BPA on pancreatic β-cells by rapidly decreasing KATP channel activity, enhancing glucose-induced [Ca^2+^] signals and insulin release. GPR30 is a non-classical membrane estrogen receptor that may participate in the insulinotropic effect of BPA on pancreatic β-cells.

**Figure 3 ijms-23-04238-f003:**
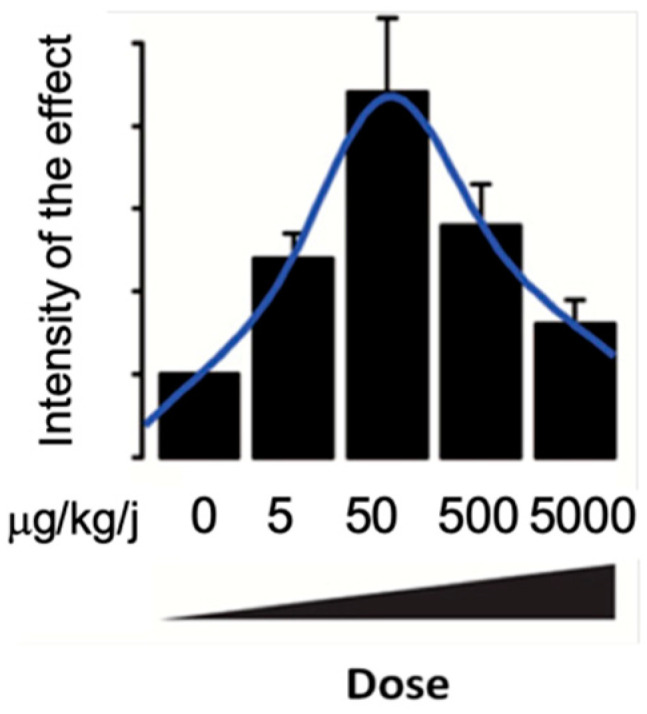
Illustration of non-monotone effects of BPA on genes involved in glucose and lipid metabolism in the liver according to Mamugi et al., 2012.

## Data Availability

Not applicable.

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
