# Peer review of "Are BPA Substitutes as Obesogenic as BPA?"

_ijms, 2022, doi:10.3390/ijms23084238_

Round 1

Reviewer 1 Report

A nice and very useful review. 

Just one question:

The authors mentioned, that the bioavailability of BPS is about 100 times higher than that of BPA. Both act as endocrine disruptors exerting estrogen-like activities. Are there any studies dealing with disturbances of the male reproductive system in offspring of rodents for example? Are there any reports about miscarriages after high BPS intake?

Author Response

We thank the reviewer for his comments. Yes there are a lot of data showing that perinatal exposure to BPS leads to disturbances of the male reproductive system in offspring of rodents (PMID: 31514588 ; PMID: 29940534), including at the transgenerational level (PMID: 31532523). They are also data reporting a link between BPS exposure and unexplained recurrent miscarriages (PMID: 34728239) or spontaneous abortions (PMID: 32702804).

These data were not included in the review which relates to the obesogenic nature of bisphenols and not to their effect on reproduction.

Reviewer 2 Report

This is a review of the diabetogenic and obesogenic effects of bisphenol A, an anthropogenic substance used in the manufacture of processed food and beverage containers.  The review does a good job of reviewing the BPA literature in terms of diabetes and obesity but several important factors are omitted.

One of the major omissions, is the lack of adequately defining a clinically relevant level of human exposure to BPA.  The authors use the terms “low dose” in describing BPA at several locations in the manuscript; however, the average human exposure level to BPA is never discussed.  The “low” in low dose has to be relative to something and that should be human exposure.  Also, the authors site in vitro studies showing 1 nM BPA causing an effect but what is the typical blood concentration of BPA?  BPA may not accumulate to any great degree in vivo so a concentration of even 1nM may not be physiologically relevant. 

Another major omission is a statement or discussion of the reality that BPA exposure is primarily from food and beverage containers; so that the more an individual eats and drinks, especially processed foods stored in plastic containers, the more BPA exposure will occur.  It is well known that eating more processed foods (tends to have high caloric content) is associated with obesity.  Furthermore, 90% of individuals who are type II diabetic are obese or overweight.  Even if energy intake is normalized, the consumption of a 500 kcal of processed food from a container lined with BPA is going to disrupt blood glucose levels differently than eating 500 kcal of fresh vegetables that never touched plastic and were picked fresh from a garden.  Other limitations were noted in individual studies in this review.  This topic needs to be addressed as a reality and potential limitation when discussing BPA exposure and obesity and/or diabetes.

Some statements were nonsensical to this reviewer:

Line 263; “Acute exposure to BPA at a dose lower than the NOAEL (1.2 mg/kg/day) can induce hepatic lesions…” If hepatic lesions are found after acute exposure to BPA at levels below the No Observed Adverse Effect Level (NOAEL) then, the study used to establish the NOAEL value is, by definition, flawed.  The NOAEL needs to be changed if that is the case because hepatic lesions would be found in those types of studies.

Line 122; the authors describe Agouti-related peptide and neuropeptide Y as “orexigenic neurons”.  These are neurotransmitters and not neuronal cell types.  Perhaps this is just a typo but it is confusing.

Minor concerns

The excessive citing of reviews in the Introduction is problematic.  References 1, 6 and 7 cited in several locations are simply reviews.  Actually, reference # 1 and 15 are the same reference which, again, is a review.  This is a review and as such should cite the original data sources as much as possible. The authors do this in the later sections of the review.

Round 2

Reviewer 2 Report

The authors did a good job in addressing the reviewers' comments.